# Chemical Composition and Polyphenolic Compounds of Red Wines: Their Antioxidant Activities and Effects on Human Health—A Review

Boris Nemzer [1,2,*], Diganta Kalita [1], Alexander Y. Yashin [3] and Yakov I. Yashin [3]

[1] Department of Research & Development, VDF FutureCeuticals, Inc., Momence, IL 60954, USA; Diganta.kalita@futureceuticals.com

[2] Department of Food Science and Human Nutrition, University of Illinois at Urbana-Champaign, Urbana, IL 61801, USA

[3] International Analytical Center of Zelinsky Institute of Organic Chemistry, 119991 Moscow, Russia; yashin@interlab.ru (A.Y.Y.); yashinchrom@mail.ru (Y.I.Y.)

\* Correspondence: bnemzer@futureceuticals.com; Tel.: +1-815-507-1427

**Abstract:** Red wine, an alcoholic beverage is composed of a spectrum of complex compounds such as water, alcohol, glycerol, organic acid, carbohydrates, polyphenols, and minerals as well as volatile compounds. Major factors that affect the levels of phenolic compounds in red wines are the variety of grapes and the storage of the wines. Among the constituents of red wine, phenolic compounds play a crucial role in attributes including color and mouthfeel and confer beneficial properties on health. Most importantly, phenolic compounds such as flavanols, flavonols, flavanones, flavones, tannins, anthocyanins, hydroxycinnamic acids, hydroxybenzoic acids, and resveratrol can prevent the development of cardiovascular diseases, cancers, diabetes, inflammation, and some other chronic diseases.

**Keywords:** red wine; polyphenols; antioxidant activity; health benefits

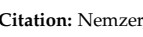



## 1. Introduction

Wine is one of the oldest traditional alcoholic beverages consumed worldwide. It was produced more than 6000 years ago in ancient Egypt and has been used as a part of the diet and in various therapies [1,2]. Wines are produced by the fermentation of grape must and can be classified as red, white, orange, or rose wine based on their color, composition, taste, and alcoholic content. Red wine has received special interest due to its pleasant taste and the cultural tradition followed in some countries [3]. It's association with the "French paradox" has prompted consumers to prefer red wine over other alcoholic beverages such as white wine, beer, or spirits [4,5]. Red wines are complex in terms of their components and processing methodologies. The characteristic composition of red wine originates from the quality of raw materials such as grapes, regimes of alcoholic fermentation, and ageing conditions [3,6]. The ageing of wine in bottles leads to transformations and evolution resulting in changes of their color and sensory characteristics. Moreover, the extent of transformation of constituents are different among all wines as it depends on their initial composition and cellar conditions [6–8]. Thus, the quality of a wine is determined by various parameters such as alcoholic concentration, polyphenolic composition, density, color, acidity, aroma, astringency, bitterness, and flavors [6,9].

A number of epidemiological studies have indicated that moderate consumption of red wine could be associated with reduced mortality and can enhance overall health status as seen through different perspectives [10,11]. Several studies have showed that moderate consumption of wine is beneficial to health because it offers protection against neurological diseases, cancers, diabetes, cardiovascular diseases, hypertension, and macular degeneration [10–15]. It has also been reported that light-to moderate wine consumption

may ensure a longer life expectancy than consuming wine in excess or none at all. However, it is known that high levels of alcoholic beverage consumption can increase blood pressure, have negative effects on the activation of the cardiovascular system and the sympathetic system, and can increase the incidence of atrial fibrillation and other health problems [11].

In this review, we provide an overview of the chemical composition of red wines and some major factors that influence these levels such as sources of raw materials and storage of wine after processing. We have also highlighted some of the in vitro and clinical studies that support the potential health benefits associated with the consumption of red wine.

## 2. Chemical Composition of Red Wine

Generally, wine is composed of water (86%), ethanol (12%), glycerol, higher alcohol, and polysaccharides (1%), organic acids (0.4%), polyphenols such as anthocyanin and tannins (0.1%), minerals, volatile compounds, and other compounds (0.5%). The major groups of chemical compounds in red wine are shown in Table 1. Following water and ethanol, glycerol is the major compound presents in wine in a range of 4–10 g/L [16]. Glycerol contributes positively to wine quality, mouth-feel, and texture properties. Organic acids are one of the common constituents of red wine. The major organic acids available in wines are tartaric, malic, citric, succinic, and acetic acid where malic and tartaric acids originate from grapes and succinic and acetic acids are produced from the fermentation process [16,17]. Grapes also contain ascorbic acid; however, it is lost during the fermentation process. Sulfites are very common chemical compounds found in all types of wine because yeast naturally produce sulfites during fermentation [16,18,19]. In ideal conditions wine should not contain more than 350 mg/L sulfite [16].

Nitrogen compounds are very commonly present in red wine since grape vines takes nitrogen from nitrate, ammonia, or urea used during the cultivation which is then converted into other amino acids for protein synthesis. Proline is found in the highest concentrations in wine because yeasts do not utilize this amino acid. Arginine is also present in high concentrations in must, but it is not found in wine at significant levels because it is consumed by yeast. Other amino acids are alanine, glutamic acid, glutamine, arginine, and γ-aminobutyric acid. Wine proteins have molecular weights ranging from of 20–40 kD. The solubility of wine proteins is highly dependent on temperature, alcohol level, and pH [16,20,21].

A class of higher alcohol (containing more than two carbons) and its derivatives of amino acid are available in red wine. For example, leucine with 3-methylbutanol, isoleucine with 2-methylbutanol, valine with 2-methylpropanol, threonine with propanol, and phenylalanine with 2-phenylethanol are present in red wine. Higher alcohols can have an aromatic effect on wines providing favorable and non-favorable aromatic profiles to the wine [16,21,22].

Terpenes are present in very small concentrations, nevertheless they have considerable impact on the organoleptic properties of grapes and wines. Over 50 terpenic compounds have been identified in grapes and wines. Some of the important terpenes include linalool, geraniol, and nerol. Terpenes in grapes and wines occur as either glycosidically bound or free volatile terpenes. Free volatile terpenes contribute to the major aroma of wine where bound terpenes do not contribute to the aroma until it is hydrolysed by acids or enzymes. Potentially volatile terpenes are two to eight times more common than free-form volatiles [23].

Among all the ingredients of red wine, polyphenols play a crucial role in its quality and aroma although only 0.1% of polyphenols are present. Most importantly, polyphenols are the major phytochemicals that are associated with the properties of red wine beneficial to human health.

Flavonoids are the most important group of compounds in red wine. The family of flavonoids consist of flavanols, flavonols, flavanones, flavones, chalcones, tannins, and anthocyanidins. The flavanols, also known as flavan-3-ols, in wines are found as catechin, epicatechin, epigallocatechin, and epicatechin 3-O-gallate in the monomeric form, and

proanthocyanidins or non-hydrolysable tannins in the polymeric form. Flavanols contribute to the color and sensory characteristics such as astringency and bitterness of wines. The levels of flavanols in red wines are constitute up to 800 mg/L whereas in white wine it ranges from 15 to 25 mg/L [6,24–26].

Flavonol is another group of compounds in the flavonoid family and comprises myricetin, quercetin, laricitrin, kaempferol, isorhamnetin, and syringetin. They exist in glycosidic forms and are linked to sugar molecules [25,27]. Flavonols also form co-pigments with the anthocyanidins and play an important role in the astringency and bitterness of wines. In red wine, the maximum content of flavonols reported was 60 mg/L [6,25,27].

Anthocyanins pigments which are composed of anthocyanidins and anthocyanins (anthocyanins are glycosides of anthocyanidin) are major water-soluble pigments present in colored grapes and the fermented product i.e., red wine. The most abundant major anthocyanins found are isomers of delphinidin, cyanidin, petunidin, peonidin, and malvinidin which exist in glycosylated and aglycosylated forms. The levels of anthocyaniandins ranges from 90 mg/L to 700 mg/L [28,29]. However, the stability of these anthocyanidins depends on various factors such as the pH, storage temperature, and raw materials used during processing.

Tannins are the unique major compounds present in red wine. The condensation of flavonols leads to the formation of tannins such as epicatechin and proanthocyanidinis. The levels of tannins increase during aging and enhance the astringency of wines that could occur via polymerization and interaction with other molecules [9]. The level of condensed tannins in red wine ranges from 1.2 to 3.3 g/L [22]. Grapes as well as wine contain some hydrolysable tannins, which are esters of gallic acid (gallotannins) and ellagic (ellagitannins) bonded to sugar molecules. The content of hydrolysable tannins could vary widely from 0.4 to 50 mg/L. The levels of proanthocyanidins are several-fold higher in red wines than in white wines. The USDA database reported that proanthocyanidin in red, rose, and white wines constituted by monomers, dimers, trimers, and polymers were up to 61.63 mg/100 g 2.20 and 0.81 mg/100 g, respectively [30,31].

Naringenin is the main compound of flavanones present in red wine and it reaches 25 and 7.7 mg/kg in red and white wines, respectively. The level of flavones ranging from 0.2 to 1 mg/L in wine [6]. Chalcones, a subclass of flavonoids which are important intermediates and are precursors of vast range of flavonoid derivatives, are found in grapes or wine.

Non-flavonoid compounds such as phenolic acids comprising hydroxybenzoic acids, hydroxycinnamic acids, and stilbenes are the major polyphenolic compounds found in grapes. These polyphenols range from 60 to 566 mg/L in red wine. Other phenolic acids available in wines are vanillic, gentisic, syringic, salicylic, and protocatechuic acids [6,24]. The gallic acid levels are upto 70 mg/L in red wine. Hydroxycinnamic acids are one of the major polyphenols in grapes and are responsible for the browning and aroma of wine. The average amount of hydroxycinnamic acids is about 100 and 30 mg/L in red and white wines respectively. Other forms of hydroxycinnamic acids such as caffeic, coumaric, sinapic, and ferulic acids are present in both red and white wines [32–34].

Stilbenes are non-flavonoids compounds present in grapes and their end products such as juice and wine. Resveratrol is one of the naturally abundant major antioxidant compounds belonging to the stilbene phenolic compounds. It has been shown that the juice of red grapes contains trans-resveratrol in the range of 0.01–1.10 ppm and cis-resveratrol in the range of 0.003–0.230 ppm. Trans-resveratrol (0.05 ppm) was detected in white wine, while cis resveratrol was not detected at all. In red wines, on average, about 3.15 ppm of trans-resveratrol and 1.84 ppm of cis-resveratrol were found. Apart from trans-resveratrol, other stilbenes reported in wines are trans-piceid, hopeaphenol, ampelosin A, isohopeaphenol, piceatannol, pallidol, e-viniferin, miyabenol C, r-viniferin, r2-viniferin. The levels of these stilbenes are reported to be low but under biotic or abiotic stress their level goes up to 100 mg/L [35–39]. The levels of cis and trans resveratrole differ among the varieties of grapes as well as the geographic location where they are grown [38]. Significant compo-

sitional difference of red and white wines from different countries in terms of resveratrol were found which is summarized in the Table 2. Tyrosol was found to be upto 45 mg/L in white wine and between 20 and 60 mg/L in red wines whereas hydroxytyrsols were found to be 3.89 mg/L in red wine [6,39,40].

**Table 1.** Major chemical compounds in red wine.

| Chemical Compounds | Contents | Reference |
|---|---|---|
| Glycerol | 4–10 g/L | [16] |
| Organic acids | upto 6 g/L | [16,17] |
| Sulfur-containing compounds: sulfite | 10–80 mg/L | [18,19] |
| Nitrogen-containing compounds: | 300 to 1300 mg/L | [20] |
| Amino acids proteins | 70–80 mg/L | |
| Higher alcohol | 300–600 mg/L | [16,21,22] |
| Isoamyl alcohol | 140–417 mg/L | |
| Minerals: potassium, nitrogen, phosphorus, sulfur, magnesium and calcium, boron, manganese, iron, and others | Total 1.5–3 g/L | [17] |
| Flavonoids: flavonols | up to 100 mg/L | [6,24] |
| Flavanols | up to 800 mg/L | [6,24] |
| Anthocyanins | up to 700 mg/L | [6,24] |
| Condensed tannins | 1.2–3.3 g/L | [24] |
| Hydrolysable tannins | up to 50 mg/L | |
| Proanthocyanidins | 1 g/L | [24] |
| Hydroxybenzoic acids | 2–500 mg/L | [16] |
| Hydroxycinnamic acids | up to 100 mg/L | [6,24] |
| Gallic acid | up to 70 mg/L | [6] |
| Stilbenes | 1.1–2.7 mg/L | [34–36] |
| Hydroxytyrosol | up to 3.89 mg/L | [6] |

**Table 2.** Level of trans- and cis-resveratrol in some red wines.

| No. | Type of Wine, Grape Variety, Year of Production, (Country of Production) | Resveratrol Content in mg/L | | |
|---|---|---|---|---|
| | | Trans-Resveratrol | Cis-Resveratrol | Total |
| 1 | Red wine, "Exposicion Carmenere", 2005, (Chile) | 1.80 | 1.20 | 3.00 |
| 2 | Red wine "Estampa Cabernet Sauvignon", 2006, (Chile) | 2.00 | 0.60 | 2.60 |
| 3 | Red wine "Estampa Cabernet Sauvignon Merlot", 2006, (Chile) | 1.60 | 0.80 | 2.40 |
| 4 | Red wine "Negroamaro Salento IGT", 2006. (Italy, Apulia) | 0.80 | 1.20 | 2.00 |
| 5 | Red wine "Merlot Myskhako", 2006, (Russia) | 0.50 | 1.40 | 1.90 |
| 6 | Red wine "Salento IGT", 2006, (Italy, Apulia) | 0.40 | 0.80 | 1.20 |
| 7 | Red wine "Vistamar Cabernet Sauvignon", 2006, (Chile) | 0.80 | 0.20 | 1.00 |
| 8 | Red wine "Cabernet Myskhako", 2005, (Russia) | 0.20 | 0.40 | 0.60 |
| 9 | Red wine "Nero d'Avola Sangiovese Emporio", 2004, (Italy, Sicily) | 0.50 | 0.10 | 0.60 |
| 10 | Red wine "Las Moras Malbec", 2006, (Argentina) | 0.25 | 0.35 | 0.60 |
| 11 | Red wine "Merlot Tamani", 2006, (Russia) | 0.40 | 0.10 | 0.50 |
| 12 | Red wine "Cabernet Tamani", 2006, (Russia) | 0.40 | 0.10 | 0.50 |
| 13 | Red semi-dry Cabernet Sauvignon, 2006, (South Africa) | 0.30 | 0.20 | 0.50 |
| 14 | Red wine Chianti Badiolo, (Italy) | 0.30 | 0.10 | 0.40 |
| 15 | White wine Estampa Chardonnay, 2006, (Chile) | 0.35 | 0.05 | 0.40 |
| 16 | Red wine "Las Moras Bonarda", 2006, (Argentina) | 0.09 | 0.15 | 0.24 |
| 17 | White wine "Chardonnay Myskhako", 2007, (Russia) | 0.12 | 0.07 | 0.19 |
| 18 | White wine "Sauvignon Blanc Myskhako", 2007, (Russia) | 0.09 | 0.02 | 0.11 |
| 19 | Rose wine "Folonari Bardolino Chiaretto", 2005, (Italy) | 0.05 | 0.06 | 0.11 |
| 20 | White wine "Malvasia Chardonnay Salento IGT", 2006, (Italy, Apulia) | 0.10 | 0.01 | 0.11 |
| 21 | White wine "Chardonnay Sicily IGT", 2005\2006 (Italy, Sicily) | 0.05 | 0.01 | 0.06 |

### 3. Analysis of Polyphenols and Other Compounds in Red Wine

More than 100 polyphenol compounds, including flavonoids and non-flavonoids, have been identified by chromatography-mass spectrometry and other analytical techniques in red wines [6]. The high content of polyphenolic compounds in red wines is associated with the fact that the fermentation process of grape juice in red wines includes the skin of berries and grapes [6]. The skin of grapes is particularly rich in polyphenol compounds. Many polyphenols are water-soluble, while others are extracted only with water–alcoholic mixtures. In the production of white wine, the skin grape berries are not used in the fermentation process, but only grape juice is used. For the analysis of wine, a whole arsenal of modern analytical methods including molecular spectroscopic, chromatographic, electrochemical, and other analytical methods are used.

#### 3.1. Spectrophotometric Methods

Based on the absorbances at 280 nm of phenolic compounds, the UV-Vis spectrophotometric method is the most common and economically viable analytical method for screening the level of phenolic compounds in red wine and its precursors. This method is appearing to be simple and inexpensive, however, the disadvantage of this technique is the interference caused by other phenolic compounds in the absorptivity of phenolic compounds [41,42]. In general, the Folin–Ciocalteu reagent method has been adopted by several investigators to quantify the total phenolics where phenolic compounds react with a mixture of phosphotungstate and phosphomolybdate reagents to form a blue-colored and show the absorbances at 760 nm. Garcia-Hernandez et al. [42] quantified total phenolics as gallic acid equivalent or three red wine varieties such as Syrah, Graciano, and Tempranillo. A number of studies have been investigated to analyse the phenolic compounds in grapes and red wines.

#### 3.2. Gas Chromatography-Mass Spectroscopy

Gas chromatography is a useful technique to quantify volatile and non-volatile compounds in red wines. Gas chromatography connected to mass spectrometer detectors is commonly used techniques to quantify phenolic compounds as well. Various methods have been adopted to preconcentrate the analytes by solid phase microextraction, headspace technique and ultrasound -assisted extraction [43–45]. Perez-Jimenez et al. [45] analysed the phenolic compounds and volatile components in six varieties of *Vitis vinifera*. Ziółkowska et al. [46], analyzed 38 white and 41 red wine samples processed from different varieties countries by solid-phase microextraction attached with mass spectrometry SPME-MS. Valentina et al. [47] investigated the authenticity of 83 South American red wines that represent a specific chemical profile of wine over another category by analyzing volatile compounds using head space solid phase micro extraction gas chromatographic spectroscopy (HS-SPME-GCMS) methods. Stupaka et al. [48] developed a rapid and simple method for alcoholic strength determination and screening of common denaturants adopting the headspace solid–phase sampling of volatiles by solid phase micro extraction (HS–SPME).

#### 3.3. High Performance Liquid Chromatography (HPLC)

HPLC is the most commonly used method to quantify phenolic compounds in red wine owing to its simplicity, reliability, high sensitivity, robustness, and repeatability. However, the methods can differ depending on the detectors used in the HPLC system [49–60]. Carneiro et al. [50] analyzed some red wines originating from Brazil and Argentina and found that vanillic acid, syringic acid, ellagic acid, quercetin, and melatonin were the major phenolic compounds quantified in different wine samples. In addition to HPLC, various new techniques such as ultra-high performance liquid chromatography (UHPLC) have been used to enhance the separation and lower the retention time of the analytes [50–56]. Lukic′ et al. [51] quantified 58 phenolic compounds in several red wine samples using ultra-performance liquid chromatography (UPLC) with a triple quadrupole

mass spectrometry (UPLC-QqQ-MS/MS) system. Del-Castillo-Alonso et al. [52] used a UPLC system to quantify phenolic acids such as quercetin, kaempferols, myricetins, and isorhamnetins in grapes and wines. Using UPLC coupled to a photodiode detector-quadrupole/time of flight-mass spectrometry (PDA-Q/TOF-MS) system, Wojdylo et al. [53] analysed the phenolic content in processed wines. Simoneetti et al. [54] determined the total content of polyphenols in 13 different types of Italian wines and reported them to be in a range of 1365–3326 mg/mL in red wines, which were 14 to 23 times higher than in the white ones. The content of flavanols in red wines was 16 to 19 times higher than in white wines. Flavonols in red wines contained about 15 mg/mL and no flavonols were found in white wines. McDonald et al. [25] systematically studied the polyphenol content in a wide variety of wines originating from different geographical locations in Italy, Chile, France, California, Australia, Bulgaria, Morocco, and Hungary. The total flavonol content of these wines varied between 4.6–41.6 mg/L, and Cabernet Sauvignon wine had higher levels of polyphenols because of the thick skins of grapes used to prepare the wine. Berente et al. [29] developed a method to quantify the anthocyanins in red wine by HPLC using 52 different types of German red wine. The color of red wines was determined by mono- and diglucosides of anthocyanins of five original anthocyanidins: delphinidin, cyanidin, petunidin, peonidin, and malvidin. De Villiers et al. [28] investigated the purple-red color of young red wine associated with anthocyanins by HPLC and HPLC-MS. Recently, some stilbenes have been quantified by UPLC-MS/MS, among them trans-piceid was the most abundant in white wine (average of 155 g/L), whereas, cis- and trans-piceids and hopeaphenol were abundantly present in red wine (average of 3.7 and 3.2 mg/L, respectively).

### 3.4. Nuclear Magnetic Resonance (NMR)

NMR (nuclear magnetic resonance) spectroscopy have been successfully used to characterize wines and its metabolites. It has also been used as a viable option for the quality assurance of foods and beverages. [60–63]. A number of metabolites belongings to amino acids, organic acids, sugars, and phenylpropanoids such as leucine, isoleucine, valine, threonine, alanine, arginine, glutamine, g-aminobutyric acid, proline, tyrosine, succinate, acetate, malate, tartarate, citrate, glucose, fructose, sucrose, 2,3-butanediol, glycerol, 2-phenylethanol, trigonelline, cis/trans-caftaric acid, cis/trans-caffeoyl malate, and cis/trans-coutaric acid have been identified by using 1H NMR spectroscopy [63]. D′Onofrio et al. [64] characterized the phenolic profiling of three Italian red wines such as Gglioppo, Magliocco, and Nerello Mascalese using NMR, HPLC/UV-Vis and spectrophotometric methods and reported that the major compounds were anthocyanins, catechin, epicatechin, gallic acid, pyrogallol, tyrosol, and 2-phenylethanol. Magliocco was the richest wine in monomeric anthocyanins (two-fold), catechins, and low molecular weight phenolics (LMWP). Barátossy et al. [65] carried out a complex chemometric analysis of red and white wine samples based on their 1H NMR spectra. Cassino et al. [62] monitored metabolite changes during storage using 1H NMR spectroscopy for 10 different red wines (*Vitis vinifera*). Forina et al. [66] identified and quantitated Malvinidin-3-glucoside (Mv3g) as the major anthocyanin and other metabolites such as catechin and epicatechin, ethyl caffeate, syringic acid, gallic acid and tyrosol metabolites by NMR.

### 3.5. Inductively Coupled Plasma Mass Spectroscopy (ICPMS)

ICPMS is considered one of the most popular techniques applied for elemental composition studies and it is based on inductively coupled plasma (ICP) as a source of excitation and ionization [67–70]. Fermo et al. [67] investigated the detected and analysed 20 elements such as K, Mg, Ca, Na, Rb, Al, Cu, Mn, Zn, Pb, As, Cd etc. by inductively coupled plasma optical emission spectrometry (ICP-OES) and ICP-MS in some Italian wines. With regards to the main elements, K was the most abundant one in all the samples followed by Mg and calcium. Kara et al. [68] studied the level of different forms of aluminum in wine samples by developing a new method for the separation and speciation analysis of aluminum and aluminum complexes (organic and inorganic) by LC–ICP–MS and ICPMS spectrometer.

Gajek et al. [69] evaluated the influence of type of wine (180 wine samples) and their origin on the levels of 28 elements such as Ag, B, Ba, Be, Bi, Ca, Cd, Co, Cr, Cu, Hg, Fe, K, Li, Mg, Mn, Mo, Ni, Pb, Rb, Sb, Sn, Sr, Te, Ti, Tl, U, Zn following the ICPMS methods. They found that in general, red wines contained higher values of B, Ba, Cr, Cu, Mn, Sr and Zn compared to other wine types (white and rose).

*3.6. Other Analyticial Techniques*

In addition to the major spectroscopic and chromatographic techniques, as mentioned above some other alternative analytical techniques have been used for specific analytes. For example, ion chromatography was used to analyze sulfites [18], capillary electrophoresis was applied to analyze resveratrol [71], electrochemical methods were used to measure total phenolics, and the antioxidant activities of red wines [72]. Gel Permeation Chromatography (GPC) was used to analyze some polymeric anthocyanins, phenolic acids, flavonoids, and carbohydrates/polysaccharides in wines [73]. Electronic devices such as electronic nose and tongue have been used for the detection of phenolic compounds in wines [74,75]. A major advantage of such electronic equipment is to avoid the use of organic solvents. Ceto' et al. [74] used an electronic nose device to analyse polyphenol compounds in red wine [64]. Rudnitskaya et al. [75] used an electronic tongue to determine the organic acids and phenolic acids in Madeira wines.

## 4. Factors Affecting the Phenolic Content

As aforementioned, the phenolic content of wine plays the most significant role in its organoleptic properties and quality. Several internal and external factors affect the levels of phenolic compounds in red wines. Among them the major factors are: (a) the variety of grapes with geographic locations and processing; and (b) the storage of wines.

*4.1. Grape Varieties*

Worldwide grapes, *Vitis vinifera* L. are recognized as one of the fruits most rich in phenolic compounds. They are used as primary resources to produce wines. Grapes are produced to the extent of 75 million tons per year, of which 70% are processed into wine, 27% are consumed fresh, and 2% are dried. The grape variety largely determines components of red wines including phenolic content, minerals, and other nutrients of red wine. High amounts of phenolic compounds are present in grapes including skin, pulp, and seeds [76–82]. Generally, red grapes contain higher levels of phenolic compounds than white grapes. However, their levels varied significantly within the varieties and large differences were observed in different varieties. The common varieties used in processing of red wines are Merlot, Cabernet Sauvignon, Tempranillo, Syrah, Pinot Noir, Sangiovese, Grenache, Malbec, Primitivo and others. Some hybrid grape cultivars, for example Isabel (*V. vinifera* × *V. labrusca*) are used for grape production [77]. The profiles of phenolic compounds vary with respect to different varieties and maturation stages of grapes. For instance, Cabernet Sauvignon, Cabernet Franc, and Pinot Noir had a higher content of phenolic compounds than other grape varieties. The content of phenolic compounds differs in different varieties. For example, Brezoiu et al. [78] observed that between the two varieties of grapes, Feteasca Negra and Cabernet Sauvignon, the Cabernet Sauvignon variety had higher levels of phenolics than Feteasca Negra. de Peredo et al. [59] similarly observed that the white grape varieties Albariño, Chardonnay, and Gewurtztraminer had lower levels of phenolic compounds than the red ones such as Cabernet Sauvignon, Graciano, Malbec, Mencía, and Merlot. However, white grapes exhibited a high content of hydroxycinnamoyl, tartaric acids, and caftaric acids. Phenolic compounds such as catechin, epicatechin, resveratrol, rutin, and quercetin were quantified as major phenolic acids in the grape varieties such as Sangiovese, Merlot, Cabernet Sauvignon, Canaiolo Nero, Colorino del Valdarno, Foglia Tonda, and Montepulciano in a study by Iacopini et al. [79]. It was found that Cabernet Sauvignon exhibited the highest level of resveratrol, Sangiovese had the highest levels of quercetin and rutin, and Colorino showed the lowest level of rutin. Several studies demonstrated differences in the con-

tents of polyphenolic compounds of grapes varieties. Guerrero et al. [80] investigated the phenolic compound profiles of red grape varieties such as Jaen Tinto, Palomino Negro, and Tintilla de Rota, Cabernet Sauvignon, and Tempranillo and they quantified several phenolic compounds sch as trans-caftaric acid, catechin, trans-coutaric acid, cyanidin 3-O-glucoside, cyanidin-3-p-coumaroylglucoside, delphinidin 3-O-acetylglucoside, delphinidin 3-O-glucoside, epicatechin, isorhamnetin 3-O-glucoside, kaempferol 3-O-glucoside, malvidin 3-O-acetylglucoside, malvidin-3-caffeoylglucoside, malvidin 3-O-glucoside, cis-malvidin-3-p-coumaroylglucoside, trans-malvidin-3-p-coumaroylglucoside, myricetin-3-O-glucuronide, myricetin-3-O-glucoside, peonidin 3-O-acetylglucoside, peonidin 3-O-glucoside, petunidin 3-O-acetylglucoside, petunidin-3-p-coumaroylglucoside, petunidin-3-O-glucoside, quercetin-3-O-rutinoside, and syringetin-3-O-glucoside. Kondrashov et al. [81] investigated the levels of total phenolics and selected vitamin and mineral contents in 10 different red wines (six Cabernet Sauvignon and four Merlot) and found that the phenolic contents were higher in Cabernet Sauvignon wines than in Merlot wines. Leeuw et al. [82] studied the level of phenolic acid (resveratrol and procyanidins B1 and B2), flavonol, anthocyanidin, flavan-3-ol among the wine prepared from seven of the grape varieties, Merlot, Syrah, Cabernet Sauvignon, Pinot Noir, San Giovese, Nero D'Avola, Malbec and Primitivo, and reported a deep variation among the varieties.

*4.2. Storage*

### 4.2.1. Ageing and Co-Pigmentation

Ageing of wine, defined as the duration of time after bottling until consumption, is a factor that is crucial for the taste and quality of wines since during the storage time phenolic compounds undergo several chemical transformations [6]. Changes or isomeric transformation or degradation/fragmentation of phenolic compounds cause an intense color and different sensory characteristics relative to the original ones. For example, some anthocyanins convert to flavanol–anthocyanin adducts, oligomers, polymeric pigments that develop different colors and tastes [83]. Phenolic compounds undergo condensation and fragmentation with increasing storage time. Older red wines have higher levels of polymeric pigments while younger wines have higher levels of anthocyanins [9]. However, the formation of polymeric pigments depends on various factors such as pH, phenolic content, oxygen permeability, and microorganisms used during the processing. The effect of aging leads to the evolution of flavonols and flavones contents in wine with aging time [84]. In a study on the effect of aging of red wines prepared from *Vitis vinifera* L. such as Tempranillo, Graciano, and Cabernet Sauvignon it was found that the level of anthocyanins decreased upto 43–66% whereas those of poranthocyanidins were increased [8]. Cassino et al. [62] analysed the composition of 10 different red wines that were bottled and stored in a wine storage cellar for 24 months. They found that there was a decrease in the levels of organic acids but an increase in the esters such as ethyl acetate and ethyl lactate. The ageing process is controlled by multiple factors, such as chemical reactions among the ingredients, storage temperature, barrels, and stoppers [6]. Zhang et al. [7] investigated the influence of grape variety, vineyard location, and grape harvest maturity, combined with different oxygen availability treatments, on red wine composition during bottle aging of wines where they reported that grape variety, vineyard location, and grape maturity had a greater influence on wine composition than bottle-aging conditions. They found that factors such as vineyard location, grape variety, and grape maturity contributed significant influences on the evolution of total and free aldehyde compounds in wine during bottle aging.

Storage temperatures play a significant role in the transformation of phenolic compounds in red wine over time which is directly related to the kinetics of the reactions that occur during the aging process. Cassino et al. [62] reported that the phenolic contents increased by 40% of their initial levels when some red wine (Barbera, Nebbiolo, Ruchè and Grignolino) were stored for 4 years at 12 °C. In a comparative study to determine the role of different temperatures, Esteban et al. [85] observed that red wines preserved at

around 15 °C contained higher anthocyanin and tannin levels compared to those preserved at temperatures around 25 °C.

It is well known that the color of red wine originates mainly due to presence of anthocyanins. Moreover, the color of anthocyanins in wine is dependent on the pH of the aqueous solution. At a lower pH, the red color of anthocyanin is due to the flavylium cation, as the pH increases the color of the anthocyanins degrades. As a result, the color of the red wines (pH 3.2−4.0) is likely to fade because the colorless hemiketals (>70%) present in such wines are in equilibrium with other forms [86]. However, instead of the color of the red wine being diminished, it is enhanced by aging and storage. This is possible due to the association of anthocyanin with other compounds (usually non-colored) present in the solution, this phenomenon is known as co-pigmentation [86]. Co-pigmentation stabilizes and enhances the color of red wine [86,87]. In nature, the potential color enhancement is fixed for a given pigment-cofactor pair and the observed color in solution depends on the concentration of pigment, the molar ratio of a cofactor to the pigment, pH, the extent of non-aqueous conditions, and the anions in solution. It appears that there should be a minimum concentration of anthocyanin available before significant co-pigmentation is detectable [86,87].

Different types of compound including alkaloids, amino acids, nucleotide, metals, and phenolic compounds act as co-pigments. Flavonoids, particularly flavanols and flavonol, and hydroxycinnamoyl derivatives, have been reported to be some promising co-pigments [86]. Flavonols are one of the major flavonoids but are weaker co-pigments than flavanols owing to their non-planar structures. Epicatechin is a better co-pigment than catechin. Some of the anthocyanin co-pigment-associated adducts occur because of direct reactions between anthocyanins and flavan-3-ols forming the dimeric-type flavanol-(4,8)-anthocyanin (F–A) and anthocyanin-(4,8)-flavanol (A–F) adducts, oenin-(O)-catechin co-pigment adducts that evolve the color of red wines [9,86,88]. The addition of phenolic acids such as caffeic acid and p-coumeric acids during the wine-making process of cabernet Sauvignon and Pinot Noir wines indicated that p-coumaric acid was a stronger co-pigment compound than caffeic acid [89].

### 4.2.2. Barrel

Wines are stored in wooden barrels after manufacturing or processing wherein some chemical compounds diffuse into the wine solution. Therefore, wood composition itself plays a critical role in the compositional changes of wines. The woods used to construct the barrels are always processed before storage. For example, toasting is a major step in preparing wood barrels to store wines. It causes significant changes to the biomolecules in the wood which in turn affects the preservation of polyphenol content in wines after storage. de Simón et al. [90] studied the effect of the toasting levels using light (115–125 °C), medium (200–210 °C), and heavy (220–230 °C) toasted wood barrels on the final composition of stored red wines. They found that higher levels of O-cresol, phenol, coniferaldehyde, sinapaldehyde and syringol with heavy toasting, which produce concentrations at least three times higher than those with light toasting. Frangipane et al. [91] investigated the levels of phenolic compounds in Merlot wine after storage in different types of wood barrel with varied roasting levels and found that heavy toasting resulted in higher levels of phenolic compounds where there was an increase in phenolic compounds except vanillin and ellagic acid. It was concluded that the heavy toasting process produced a higher evolution of phenolic compounds in the wine. In addition, Dumitriu et al. [92] reported that lightly toasted wood barrels restored higher level of phenolic compounds. In another study, Watrelot et al. [93] did not observe significant changes in the level of anthocyanins in Cabernet Sauvignon wine after storage in wood barrels with different toasting levels.

### 4.2.3. Closure

A closure acts as a permeable membrane that allows the exchange of oxygen between wine and the outside environment after bottling and is considered a key factor during

aging in the quality of wines. Depending on the type of closures used, oxygen can enter the bottle at different oxygen transmission rate (OTR) [94]. The oxygen that enters the bottle could affect the oxidation of wine materials resulting in some off-odors. Various commercially available closures used in the bottling are natural cork, colmated natural cork, agglomerated cork, synthetic cork, and screw caps. Several studies indicated that cork or synthetic closures are more permeable to oxygen than screw caps and thus contribute to the intense color and astringency.

Baiano et al. [95] investigated the oxygen permeability of closures on chemical, physical, and sensory qualities of Nero di Troia wines using synthetic stoppers that had different low, medium, and high rate of oxygen permeability. The results indicated that oxygen permeating through the medium and high oxygen permeable stoppers favored the increase in polymerization reactions within tannins and between flavan-3-ols and anthocyanins. de Esteban et al. [85] studied the influence of different stoppers on the evolution of phenolic compounds during the aging process of Malbec wine. Vidal et al. [96] investigated the effect of different closure using four screw caps, two synthetic and two technical corks on the shelf lives of Merlot/Tannat red wines. They observed that unlike closures with the highest oxygen transmission rate (OTR), the two technical corks and the two screw caps with a Saranex seal, having the lowest OTR matched with the wines exhibiting a low total $O_2$ content at equilibrium (from 4th to 18th month), with more free $SO_2$ and a less changed color. In addition, Wirth et al. [97] studied the impact of oxygen transfer rates on the phenolic composition of Grenache red wines using synthetic closures with controlled oxygen permeability and it was concluded that exposure to oxygen has an important influence on the oxidation of some phenolic compounds, especially anthocyanins. Rossetti et al. [98] investigated the effect of different types of stoppers made of natural cork, blends of natural cork and polymers, agglomerated natural cork, and technical cork 1 + 1 on the phenolic and volatile composition of four Italian red wines, namely Merlot, Lagrein red, Lagrein rosé, and St. Magdalener) stored in bottles for 12 months. The resulting changes in phenolic and volatile concentrations were presumably due to non-oxygen-mediated reactions occurring during 12 months of storage in bottles.

## 5. Antioxidant Activity of Red Wines

Several works in the literature demonstrated the antioxidant activities of red wines. Various analytical techniques have been adopted for in vitro and in vivo studies to characterize the antioxidant activity of red wines. For example, free radical scavenging activity assays such as 2,2-diphenyl-1-picrylhydrazyl (DPPH), ABTS (2,2′-azino-bis(3-ethylbenzothiazoline-6-sulfonic acid)), FRAP (ferric-reducing antioxidant power), ORAC (oxygen radical absorbance capacity), chemiluminescence, HPLC, spectrophotometry, and amperometry have been used to investigate the in-vitro antioxidant activities [99–103]. It has been proposed that the antioxidant capacities of red wine were exhibited due to the presence of various antioxidants like anthocyanins, procyanidins and pyroanthocyanidins, phenolic acids catechin, epicatechin, rutin, quercetin, myricetin and others as mentioned above [104–108]. Also, it has been reported that higher polyphenols are available in red wine and contributes higher antioxidant activities compared to white wines. The polyphenols content in red wine is significantly higher than in white wine, in some cases ten times or more. Vinson et al. [109] conducted a comparative study of red and white wine-based phenol index where they found that red wine had a better phenol index than the white wines. Another study was conducted based on (+) catechin using DPPH, FRAP and BCB (beta carotene bleaching) assays, where it was reported that the average antioxidant activity of red wines was also 10 times higher than that of white wines due to the higher levels of catechin, and other flavonoid compounds. Simonetti et al. [54] investigated 13 different Italian wines for their polyphenol content and their contribution to the total antioxidant capacity. It was found that red wines contained appreciable amounts of flavonols (average 15.3 mg/L), with quercetin and rutin being the most abundant, followed by myricetin, kaempferol, and isorhamnetin accounting for only 0.7−3% of total antioxidant activities (TAA).

The effectiveness of red wine for antioxidant activity has been studied worldwide with respect to their country of origin such as USA, France, Italy, Spain, Portugal, Serbia, Croatia, Montenegro, Poland, Slovakia and Austria, Czech Republic, Greece, Turkey, Georgia, India, China, Chile, Argentina, Brazil, South Africa [54,110–122].

## 6. The Effect of Moderate Consumption of Red Wine on Human Health

A large body of literature has reported the health benefits of moderate consumption of red wine. This was prompted by the phenomenon of the "French paradox" in the nineties. In recent years, several studies have reported on the relationship between moderate red wine consumption and human health [123,124]. Over the last decades several human and animal studies demonstrated the relationships between red wine polyphenols and health status protecting from cardiovascular disease, Alzheimer's disease, breast cancer, atherosclerosis, lipid peroxidation, and inflammatory diseases [123–133]. In a recent study, Gambibi et al. [130] reported that moderate wine consumption increases the expression of key longevity-associated genes like p53, sirtuin1, catalase, and superoxide dismutase in humans. Huang et al. [131] showed that moderate alcohol consumption was associated with slower high density lipoprotein (HDL)-cholesterol decreases. Resveratrol and flavonoid compounds of red wine showed a positive effect in preventing coronary heart disease and offered other cardioprotective activities by changing the lipid profiles, reducing the insulin resistance, reducing the oxidative stress of low-density lipoprotein cholesterol (LDL-C), increasing nitric oxide (NO) bioactivity [123,124]. Red wine also promoted the reduction of blood pressure [129] and reduced oxidation of low-density lipoproteins [128]. Various studies demonstrated that moderate red wine consumption reduces the risk of serious cancer diseases of human organs such as esophagus, stomach, intestines, liver, and pancreas due to the bioactivities of phytochemicals such as lignans, quercetin, resveratrol, and flavonoids [132,134]. Moderate consumption of red wine is associated with the lowering of glucose levels and decreasing in plasma insulin in type 2 diabetes patients [135]. Several studies supported that moderate consumption of red wine is associated with a lower risk of developing neurodegenerative diseases such as Alzheimer's, Parkinson's diseases, and dementia [136–140]. Various mechanisms have been proposed for this neuroprotective effect that include inhibition of tau and β-amyloid aggregation, activation of brain-derived neurotrophic factor (BDNF), and an increase in insulin-like growth factor-I(IGF-I) [136–140]. A list of other diseases, the risk of which are reduced by the consumption of red wine is given in Table 3.

**Table 3.** Health benefits of moderate consumption of red wine.

| Name of Diseases | Positive Effects | References |
|---|---|---|
| Heart disease | Prevents heart disease, inhibit platelet aggregation | [123] |
| Oxidation of low-density lipoproteins | Inhibits low density lipo-proteins (LDL) oxidation | [128] |
| Oncological diseases | Reduces esophageal cancer, reduced the progression of malignant phase of cancer, inhibited carcinogenesis with pleiotropic effect, in vivo hepatoprotective effects, suppresses the proliferation of anchorage-independent b(4) production | [13,132,133] |
| Type 2 diabetes | Ameliorates diabetic oxidative status, lowers glucose levels | [127,141] |
| Neurodegenerative diseases: Alzheimer's and Parkinson's | Neuroprotection | [136–140] |
| Inflammatory processes | Inhibits phosphorylation activation, prevents radical formation and their activities, prevents aortic lipid deposition | [142] |
| Kidney diseases | Kidney antioxidant defenses, exert protective effects against renal ischemia/reperfusion injury, inhibit apoptosis of mesangial cells | [143] |
| Oxidative stress | Prevents radical formation, antioxidant activities | [144–146] |
| Antidiabetic activities | Controls glucose levels, prevent diabetes | [10,147] |
| Allergic diseases | Inhibits immunoglobulin (IgE) synthesis, activate of mast cells and basophils or other inflammatory cells, produce inflammatory mediators, including cytokines | [148] |

## 7. Conclusions

Red wine is an alcoholic beverage associated with diverse chemical compounds including polyphenols such as anthocyanins, flavanol, flavonol, tannins, and non-flavonoid compounds, phenolic acids and resveratrol. The spectrum of phenolic compounds influences the organoleptic properties of the wine. With ageing and proper storage conditions, red wine acquires an attractive color and sensory properties. Numerous scientific studies in recent decades have confirmed the health effects of red wine and showed that regular moderate consumption of red wine can be used to prevent cardiovascular, oncological, neurodegenerative diseases, type 2 diabetes, and other chronic diseases. Red wine suppresses the oxidative stress precursors of many diseases and can be an important component of antioxidant therapy. Moreover, it improves the intestinal microbiota leading to a healthier human body system. However, further evaluation and clinical studies are needed in this promising fields for better understanding of the therapeutic effects exerted by polyphenols, including synergistic interactions among themselves or with other dietary bioactive components. It is well established that the effectiveness of polyphenolic compounds depends on their bioavailability to perform biological action in the human body. Wine polyphenols undergo marked metabolism during their passage through the digestive system. Several biotransformations occur during this metabolism starting in the oral cavity and ending

in the gut. However, very limited research has been undertaken to date regarding the bioavailability of polyphenols in red wine. The phenolic metabolites originating from polyphenol parent molecules are of great interest to investigators and this area needs to be explored further.

**Author Contributions:** Conceptualization: B.N.; Writing: D.K., A.Y.Y., Y.I.Y.; Reviewing and editing: B.N., D.K., A.Y.Y., Y.I.Y. All authors have read and agreed to the published version of the manuscript.

**Funding:** There is no external funding for this research.

**Institutional Review Board Statement:** Not applicable.

**Informed Consent Statement:** Not applicable.

**Data Availability Statement:** Not applicable.

**Conflicts of Interest:** The authors declare no conflict of interest.

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
