# Peer review of "Chemical Composition and Polyphenolic Compounds of Red Wines: Their Antioxidant Activities and Effects on Human Health—A Review"

_beverages, doi:10.3390/beverages8010001_

Round 1
Reviewer 1 Report
BEVERAGES-1500032
The manuscript Chemical Composition of Red Wines, Antioxidant Activity, and Their Effects on Human Health: A Review, reports and discusses the chemical composition of red wines and some major factors affecting their phenolic content, such as sources of raw materials and storage of wine after processing. The authors have also included a few studies about antioxidant activity of red wine, as well as their potential health benefits.
From my point of view, the novelty of this review is only referred to the phenolic composition of red wine and major factors that influence the polyphenols levels. Nevertheless, this review does not explain properly the activities and potential bioactivities of wine phenolic compounds. Over last decade, several research groups have widely reviewed the benefits associated with the moderate consumption of red wine on human health. Therefore, new evidence to support this topic addressed should be included. In my opinion, the approach of polyphenols analysis section is unacceptable; it was not properly performed and should be deleted.
In order to further improve the manuscript, abstract and conclusions also need to be revised, as both should state briefly the main points of all sections. Furthermore, the review title should be more concise as the review is about phenolic composition of red wines and some factors affecting it
Author Response
Comments and Responses to the Reviewer
However, some minor points of criticism and questions have to be clarified and/or corrected prior publication:
LINE 24: There is also “orange wine”. Therefore, I suggest“…could be classified based on their color, composition …”.
We modified it.
LINE 28: I would switch the the word “ingredients” for “compounds” or “components” (the same for LINE 165).
We have changed it.
LINE 31: Not necessarily in “improvement”. I would remove this word (“…in changes of their color and sensory characteristics”.
We modified it.
LINE 38: I would say “could be associated” and not “is associated”.
We changed it.
LINE 52: I would switch “carbohydrates” by “polysaccharides and oligosaccharides”.
LINE 53: Actually tannins are polyphenol. I would recommend “polyphenols such as anthocyanins and tannins”.
We changed it.
LINE 67: “Proline” and not “proline “ (typing mistake).
We changed it.
LINE 105. I think it would be pertinent to explain difference between “anthocyanidins” and “anthocyanins”…
We explained it in the text.
LINE 113: “The levels of tannins increase during aging and enhance the astringency of wines.” This point must be, if true, absolutely clarified.
We have explained with supporting reference.
LINES 119-120: “61.6”, “2.2” and “0.8”, in order to better read.
We have changed it.
LINES 119-120 and 123: In my opinion, it is strange to express it in “mg/kg” instead of “mg/L”, keeping in mind that authors write about wine.
We have used the same unit as the USDA database used.
Table 1: Why sometimes is “mg/l” and other times “mg/L”? Is it a typing mistake?
We have corrected it.
LINE 140 “in the range of 0.01-1.10 ppm”.
We have corrected it.
LINE 141: “…in the range of 0.003-0.230 ppm”.
We have corrected it.
LINE 144: Vitis vinifera (in cursive)
We have corrected it.
LINES 156 and 159: Some reference?
We have inserted references
LINE 172: “Ciocalteu”
We have corrected it.
LINE 175: “Garcia” (and not Garcie)
We corrected it.
LINE 176: What about this “viz.”? I don’t understand what is its meaning.
We have changed it.
LINE 179: Why “chromatography” but “Spectroscopy”? It would write it like Gas Chromatography. And I would add “GC” in order to abbreviate in the following paragraphs (the same for Mass Spectrometry” and MS).
We have corrected it.
LINE 184: “Jimenez” (not “Jemenez”).
LINE 185 “Vitis vinifera” (in cursive, second word not in capital letters, and without hyphen.
We have corrected it.
LINE 186 or LINE 187: Indicate that this abbreviation corresponds with this technique “High Performance Liquid Chromatography (HPLC)…
We have corrected it.
LINE 193: I don’t understand why in line “186” all words of the technique is written in capital letters and not in line 193…
We have corrected it.
LINE 194: In bibliography, authors write Luki´c. I think it must be a typing miskate (in bibliography, in text…, or both of them).
We have corrected it.
LINE 195: As previously mentioned, the meaning of MS has not been indicated…
We have updated it
LINE 196: DAD? TOFII? What it means? Authors must indicate the meaning of these abbreviations (not all readers are necessarily familiarized with this abbreviations).
We have explained it.
LINE 199: The same thing in the case of “PAD”.
We have fixed it.
LINE 199: Simonetti (and not Sinometti). (just typing mistake)
We corrected it.
LINE 202: I think that the word “with” should be removed.
We corrected it.
LINE 213: De Villiers (and not De Villers).
We corrected it.
LINE 217: 3.7 and 3.2, in order to a best read.
We corrected it.
LINE 221. “spectroscopy. Ten different…”
We corrected it.
LINE 222, 237 and 296: Vitis vinifera (in cursive)
We corrected it.
LINES 219-223. I think the results concerning NMR should be more largely explained: what metabolites? And what about polyphenols and NMR?
We have expanded this section by discussing relevant studies.
LINES 227 and 228. Why Ceto et al. have their publication year (2012) in the text, but not Rudnitskaya et al.? (transposable to the rest of the manuscript…)
We corrected it.
Table 3: Analyte (and not analyte). And β-glucose (with hyphen).
We corrected it.
LINE 229. Madeira (and not Madmadeira) (typing mistake)
We corrected it.
LINE 234: What about “terroir” or region factor? Is there no impact on wine polyphenols? Is it the same composition for a Cabernet Sauvignon wine from Bordeaux and from Australia?
We have mentioned about the effect of geographic locations on the wine compositions in the text
LINE 246 and 281: Sangiovese (and not San Giovese).
We corrected it.
LINE 247: I would incorporate “Grenache” and I would remove Nero d’Avola (great grape to elaborate great wines…but not very expanded in surface or production terms around the world.
LINES 247-248: V.vinifera and V. labrusca must be written in cursive.
We corrected it.
LINE 250: “Cabernet Sauvignon, Cabernet Franc and”.
LINE 251: Not “in in” but “in” (typing mistake)
We corrected it.
LINE 251: what difference between “varieties” and “cultivars”. If authors write this phrase like this, they should explain differences between these two terms (frequently used as synonyms…)
We have changed it.
LINE 255: I would use “had” instead “have” in order to homogeneity.
We corrected it.
LINE 275: “and syringetin-3-O-glucoside” (“and” should be added to a better read).
We corrected it.
LINE 289-293 and 317-324: Some references?
We have updated it.
LINE 303: And which were the results obtained by Zhang et al.? Some conclusion?
We modified it.
LINE 357: “an increase” (not “a inecrease”) (typing mistake)
We corrected it.
LINE 368-370. This phrase should be revised and rewritten. Not well-comprehensible.
We modified it.
LINE 380. “De Esteban” or “de Esteban”? And what is their real reference number, 74 or 75?
We corrected it.
LINE 401: DPPH, ABTS, FRAP, ORAC? What are their meanings? And why in this list there are free radicals numbers…and techniques? This phrase must be corrected. Moreover, here authors could use “HPLC”.
We corrected it.
LINE 419: What is the meaning of TAA?
We corrected it.
LINE 423: Not in USA?
We corrected it.
References 47, 71, 73: Name authors should be respectfully well-written.
We corrected it.
In general, some point about sensorial analysis and polyphenols were missed.
We have mentioned about the relationships in the text, but the detail was not covered in this review.

Reviewer 2 Report
In their Manuscript entitled "Chemical Composition of Red Wines, Antioxidant Activity, and Their Effects on Human Health: A Review" for the journal Beverages, authors have reported an original, complete and qualitative work and I think that this study is really valuable and very interesting.
However, some minor points of criticism and questions have to be clarified and/or corrected prior publication:
LINE 24: There is also “orange wine”. Therefore, I suggest“…could be classified based on their color, composition …”.
LINE 28: I would switch the the word “ingredients” for “compounds” or “components” (the same for LINE 165).
LINE 31: Not necessarily in “improvement”. I would remove this word (“…in changes of their color and sensory characteristics”.
LINE 38: I would say “could be associated” and not “is associated”.
LINE 52: I would switch “carbohydrates” by “polysaccharides and oligosaccharides”.
LINE 53: Actually tannins are polyphenol. I would recommend “polyphenols such as anthocyanins and tannins”.
LINE 67: “Proline” and not “proline “ (typing mistake).
LINE 105. I think it would be pertinent to explain difference between “anthocyanidins” and “anthocyanins”…
LINE 113: “The levels of tannins increase during aging and enhance the astringency of wines.” This point must be, if true, absolutely clarified.
LINES 119-120: “61.6”, “2.2” and “0.8”, in order to better read.
LINES 119-120 and 123: In my opinion, it is strange to express it in “mg/kg” instead of “mg/L”, keeping in mind that authors write about wine.
Table 1: Why sometimes is “mg/l” and other times “mg/L”? Is it a typing mistake?
LINE 140 “in the range of 0.01-1.10 ppm”.
LINE 141: “…in the range of 0.003-0.230 ppm”.
LINE 144: Vitis vinifera (in cursive)
LINES 156 and 159: Some reference?
LINE 172: “Ciocalteu”
LINE 175: “Garcia” (and not Garcie)
LINE 176: What about this “viz.”? I don’t understand what is its meaning.
LINE 179: Why “chromatography” but “Spectroscopy”? It would write it like Gas Chromatography. And I would add “GC” in order to abbreviate in the following paragraphs (the same for Mass Spectrometry” and MS).
LINE 184: “Jimenez” (not “Jemenez”).
LINE 185 “Vitis vinifera” (in cursive, second word not in capital letters, and without hyphen.
LINE 186 or LINE 187: Indicate that this abbreviation corresponds with this technique “High Performance Liquid Chromatography (HPLC)…
LINE 193: I don’t understand why in line “186” all words of the technique is written in capital letters and not in line 193…
LINE 194: In bibliography, authors write Luki´c. I think it must be a typing miskate (in bibliography, in text…, or both of them).
LINE 195: As previously mentioned, the meaning of MS has not been indicated…
LINE 196: DAD? TOFII? What it means? Authors must indicate the meaning of these abbreviations (not all readers are necessarily familiarized with this abbreviations).
LINE 199: The same thing in the case of “PAD”.
LINE 199: Simonetti (and not Sinometti). (just typing mistake)
LINE 202: I think that the word “with” should be removed.
LINE 213: De Villiers (and not De Villers).
LINE 217: 3.7 and 3.2, in order to a best read.
LINE 221. “spectroscopy. Ten different…”
LINE 222, 237 and 296: Vitis vinifera (in cursive)
LINES 219-223. I think the results concerning NMR should be more largely explained: what metabolites? And what about polyphenols and NMR?
LINES 227 and 228. Why Ceto et al. have their publication year (2012) in the text, but not Rudnitskaya et al.? (transposable to the rest of the manuscript…)
Table 3: Analyte (and not analyte). And β-glucose (with hyphen).
LINE 229. Madeira (and not Madmadeira) (typing mistake)
LINE 234: What about “terroir” or region factor? Is there no impact on wine polyphenols? Is it the same composition for a Cabernet Sauvignon wine from Bordeaux and from Australia?
LINE 246 and 281: Sangiovese (and not San Giovese).
LINE 247: I would incorporate “Grenache” and I would remove Nero d’Avola (great grape to elaborate great wines…but not very expanded in surface or production terms around the world.
LINES 247-248: V.vinifera and V. labrusca must be written in cursive.
LINE 250: “Cabernet Sauvignon, Cabernet Franc and”.
LINE 251: Not “in in” but “in” (typing mistake)
LINE 251: what difference between “varieties” and “cultivars”. If authors write this phrase like this, they should explain differences between these two terms (frequently used as synonyms…).
LINE 255: I would use “had” instead “have” in order to homogeneity.
LINE 275: “and syringetin-3-O-glucoside” (“and” should be added to a better read).
LINE 289-293 and 317-324: Some references?
LINE 303: And which were the results obtained by Zhang et al.? Some conclusion?
LINE 357: “an increase” (not “a inecrease”) (typing mistake)
LINE 368-370. This phrase should be revised and rewritten. Not well-comprehensible.
LINE 380. “De Esteban” or “de Esteban”? And what is their real reference number, 74 or 75?
LINE 401: DPPH, ABTS, FRAP, ORAC? What are their meanings? And why in this list there are free radicals numbers…and techniques? This phrase must be corrected. Moreover, here authors could use “HPLC”.
LINE 419: What is the meaning of TAA?
LINE 423: Not in USA?
References 47, 71, 73: Name authors should be respectfully well-written.
In general, some point about sensorial analysis and polyphenols were missed.
Finally, I congratulate the authors for this interesting and very valuable work.
Author Response
Comments and Responses to the reviewer
Nevertheless, this review does not explain properly the activities and potential bioactivities of wine phenolic compounds. Over last decade, several research groups have widely reviewed the benefits associated with the moderate consumption of red wine on human health. Therefore, new evidence to support this topic addressed should be included.
- We have modified the section 6: The effect of moderate consumption of red wine on human health with some more recent studies and explanations.
Line-499-532
In my opinion, the approach of polyphenols analysis section is unacceptable; it was not properly performed and should be deleted.
- We have reorganized section 3: Analysis of polyphenols and other chemicals in red wine. We have incorporated more supporting studies including GCMS, NMR, ICP-MS, and other analytical methods.
Line 168-297.
In order to further improve the manuscript, abstract and conclusions also need to be revised, as both should state briefly the main points of all sections.
- We have modified the abstract and conclusion accordingly.
Furthermore, the review title should be more concise as the review is about phenolic composition of red wines and some factors affecting it
- We have modified the title.

Round 2
Reviewer 1 Report
The paper was revised In accordance to the reviewer´s recommendationn and it is now ready to be published